

# Applying a deep learning pipeline to classify land cover from low-quality historical RGB imagery

Harold N. Eyster and Brian Beckage

Department of Plant Biology, University of Vermont, Burlington, VT, United States of America
Gund Institute for Environment, University of Vermont, Burlington, VT, United States of America

## ABSTRACT

Land use and land cover (LULC) classification is becoming faster and more accurate thanks to new deep learning algorithms. Moreover, new high spectral- and spatial-resolution datasets offer opportunities to classify land cover with greater accuracy and class specificity. However, deploying deep learning algorithms to characterize present-day, modern land cover based on state-of-the-art data is insufficient for understanding trends in land cover change and identifying changes in and drivers of ecological and social variables of interest. These identifications require characterizing past land cover, for which imagery is often lower-quality. We applied a deep learning pipeline to classify land cover from historical, low-quality RGB aerial imagery, using a case study of Vancouver, Canada. We deployed an atrous convolutional neural network from DeepLabv3+ (which has previously shown to outperform other networks) and trained it on modern Maxar satellite imagery using a modern land cover classification. We fine-tuned the resultant model using a small dataset of manually annotated and augmented historical imagery. This final model accurately predicted historical land cover classification at rates similar to other studies that used high-quality imagery. These predictions indicate that Vancouver has lost vegetative cover from 1995–2021, including a decrease in conifer cover, an increase in pavement cover, and an overall decrease in tree and grass cover. Our workflow may be harnessed to understand historical land cover and identify land cover change in other regions and at other times.

## INTRODUCTION

Classifying land and water into different cover types or classes can produce datasets key for understanding a wide variety of phenomena ranging from equity of park systems (*Ibes, 2015*) to urban temperatures (*Zhou, Huang & Cadenasso, 2011*) and the vulnerability of birds to climate change (*Jarzyna et al., 2016*). However, the scale of land use and land cover classifications (LULC; hence land cover for brevity) overwhelm manual annotations and has required the development of automated techniques (*Risojević, Momić & Babić, 2011*). Such techniques have sought to match, and even surpass, the accuracy of manual classification (*Carrasco et al., 2019*). Artificial intelligence and machine learning

Corresponding author
Harold N. Eyster,
haroldeyster@gmail.com

methods—in particular deep learning algorithms—have increased the accuracy of land cover classifications (*Talukdar et al., 2020*; *Zhao et al., 2023*). Deep learning algorithms use artificial neural networks trained on images and their associated annotations to create models that classify new images into land cover classes (*Taghanaki et al., 2020*). Annotations indicate which part of each image belongs to which class and are used to train the model. This neural network image classification process, known as semantic segmentation, has become widely used for many tasks, including identifying landscape features for autonomous vehicles (*Cakir et al., 2022*), diagnosing diseases from medical images (*Goceri & Goceri, 2017*), and land cover classification (*Ayhan & Kwan, 2020*). Semantic segmentation, as opposed to other methods, is particularly appropriate for land cover classification because it classifies every single pixel, thus creating a map of land cover that designates every pixel as a land cover type (*Tzepkenlis, Marthoglou & Grammalidis, 2023*).

As image classification technology has advanced, so has the available data. Hyperspectral and multispectral satellite imagery is becoming widely available, and can provide resolution of less than 2 m (*Vali, Comai & Matteucci, 2020*; *Shahfahad et al., 2023*). Similarly, autonomous drones can produce very high spatial and spectral resolution for specific regions (*e.g.*, spatial resolution less than 0.02 m; *Nezami et al., 2020*). LiDAR surveys are becoming increasingly widespread, frequent, and publicly available, and can greatly enhance the identification of land cover (*Yan, Shaker & El-Ashmawy, 2015*).

However, these new datasets are insufficient for determining land cover change, which is key for understanding environmental and social systems (*Gaur & Singh, 2023*). For example, land cover change can help to identify the role of urban sprawl on bird diversity (*Montero et al., 2021*) and the effects of racist housing and lending practices on exposure to extreme heat events (*Chen, 2022*). Addressing today's challenges requires not only characterizing modern land cover, but also historical land cover, where imagery is often sparse and low-quality.

Yet land cover analyses have focused on classifying high-quality modern imagery, not lower-quality historical imagery (*Yuan, Shi & Gu, 2021*), making it unclear what analysis pipelines might be suited to classifying such historical imagery. Moreover, while much modern, high-resolution annotated imagery exists, such historical annotated imagery is rare—for example, high-resolution annotations of Metro Vancouver, BC, Canada land cover only go back to 2014 (https://open-data-portal-metrovancouver.hub.arcgis.com/, accessed 2024 Feb. 16). This lack of proven approaches to classify historical land cover, the lower quality of historical imagery, and the scarcity of annotated historical land cover make classifying historical land cover difficult.

A new deep learning neural network architecture may offer a route towards historical image classification. Recent papers have shown that a deep neural network for semantic segmentation, DeepLabv3+ (*Chen et al., 2018*), outperforms other architectures in classifying land cover (*Du et al., 2019*; *Su & Chen, 2019*; *Ayhan & Kwan, 2020*). For example, *Du et al. (2019)* compared the performance of U-Net, PspNet, SegNet, DeepLabv2 (DLv2), and DeepLabv3+ (DLv3+) in classifying land cover from RGB images. Using OA, Kappa, and F1-scores, they showed that DeepLabv3+ achieved much higher success in

nearly all cases, including an average Kappa coefficient of 0.8, relative to 0.72, 0.67, 0.68, and 0.72 for U-Net, PspNet, SegNet, and DLv2, respectively. These results, and others (*Su & Chen, 2019*; *Ayhan & Kwan, 2020*; *Chen et al., 2018*) suggest that DeepLabv3+ may offer promise for classifying historical imagery.

This superior performance of DeepLabv3+ is likely due to a number of features. First, it uses an atrous (also known as dilated) deep convolutional neural network (*Chen et al., 2018*). This feature enables it to integrate information across the whole image to make inferences about a particular part of the image (*Chen et al., 2018*). Second, it uses atrous spatial pyramid pooling (ASPP) which enables the integration of information at multiple scales (*Chen et al., 2018*). Third, it uses batch normalization (*Ioffe & Szegedy, 2015*) to greatly speed up the training time (*Chen et al., 2018*). Finally, it uses an encoder–decoder structure to refine class boundaries (*Chen et al., 2018*). Moreover, atrous convolutional and atrous spatial pyramid pooling methods may help take advantage of more of the information in historical imagery, rather than just the information in a part of the image, thereby optimizing the utility of the limited information contained in historical imagery.

However, despite the proven success of DeepLabv3+ in classifying land cover, its capacity to classify low quality historical imagery into many land cover classes remains untested. Although *Su & Chen (2019)* tested DeepLabv3+ on older images, they used high-quality Google Earth images and these images were only classified into broad land classes (*e.g.*, 'urban land' without distinguishing between lawns, buildings, roads, or types of trees *etc.*). Indeed, DeepLabv3+ has not been tested on its capacity to characterize specific vegetation types like broadleaf *vs.* coniferous trees. Moreover, previous DeepLabv3+ land cover analysis code has not been made publicly available (*Du et al., 2019*; *Su & Chen, 2019*; *Ayhan & Kwan, 2020*), limiting the potential uptake of these methods.

Here, we apply DeepLabv3+ to classify historical, low-quality RGB aerial imagery into fine-scale land cover classes and test this application with a case study of Metro Vancouver, Canada. Our article contributes to the analysis of remote sensing data by showing that fine-scale tree classes can be accurately distinguished in low quality RGB images by using a small annotated region of the historical image to fine tune a modern classification model.

## METHODS

### Pipeline overview

Our objective was to apply a deep learning pipeline to classify low-quality RGB imagery. Our pipeline (Fig. 1) used modern annotated and RGB images to train an initial DeepLabv3+ model. Then, after manually annotating a small selection of a historical RGB image, the model was fine-tuned by training it on this small annotated region. Our pipeline produced two models (a model for classifying modern land cover and a model for classifying historical land cover). These models can be used to estimate modern and historical land cover, which in turn can be compared to calculate land cover change. We applied this land cover pipeline to classify land cover in Metro Vancouver, BC for 1995. Custom code used in this analysis can be found on DOI: 10.17605/OSF.IO/BWJKN.

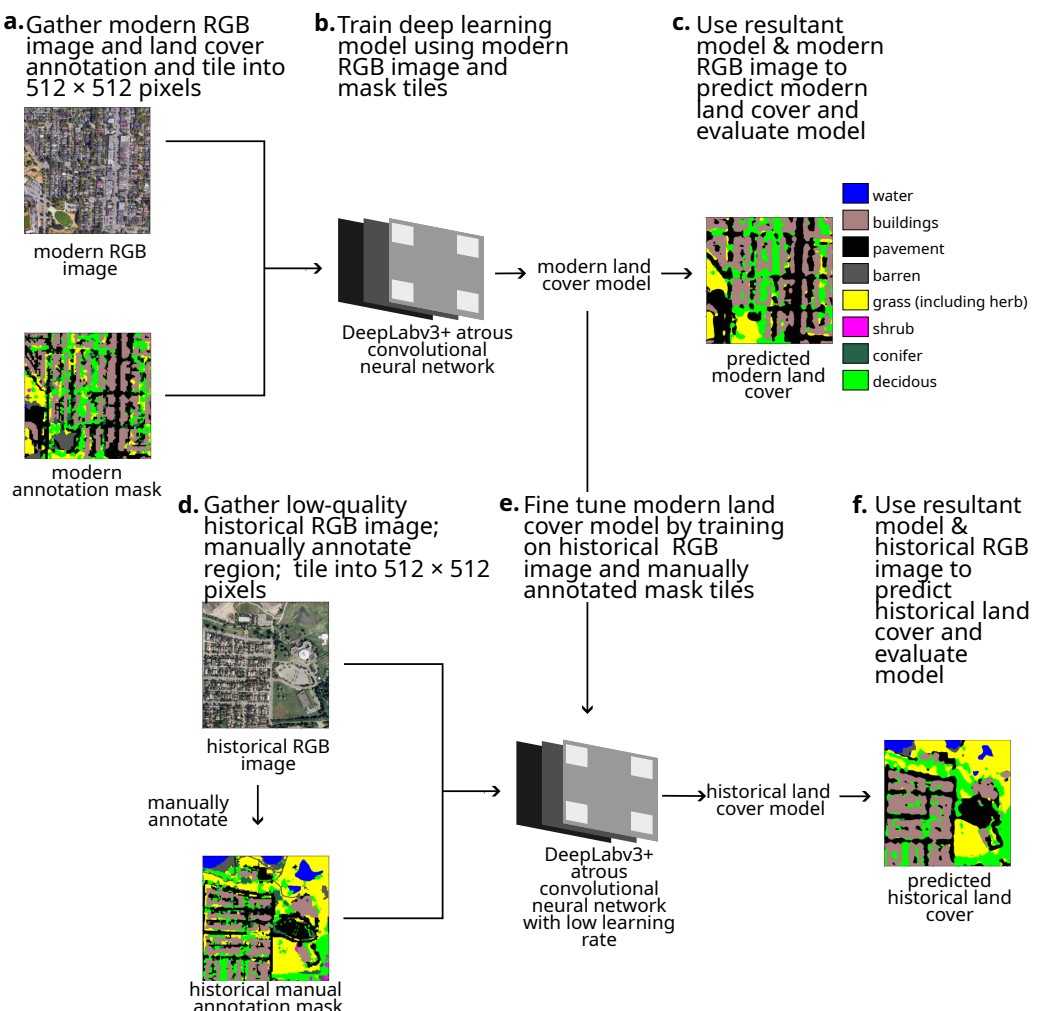

**Figure 1** **Our pipeline for using deep learning models to classify low-quality historical imagery, showing each step.** Map imagery Google, Maxar Technologies.

## Images

We used a modern, geo-referenced, annotated land cover mask and two geo-referenced RGB images—one modern and one historical (Figs. 1A, 1D). Each pixel value in the mask indicates the land cover type of the location associated with that pixel. While high-spatial resolution multispectral images are widely available now, aerial RGB images often provide the highest spatial resolution remote sensing of land in the 1990s necessary for understanding fine-scale changes in land cover such as single trees.

For our modern RGB image, we obtained modern satellite imagery taken over Metro Vancouver in 2021 from Maxar Technologies *via* Google Imagery (RGB, ∼0.5 m spatial resolution, 8 bit unsigned integer, accessed 01 November, 2022) and downsampled to 1 m spatial resolution (Fig. S6 for map of training imagery extent). We cropped the modern mask and RGB image to cover an area of 48,640 m × 28,672 m. We obtained a 5 m spatial resolution annotated land cover mask for Metro Vancouver representing 2019 (*Metro*

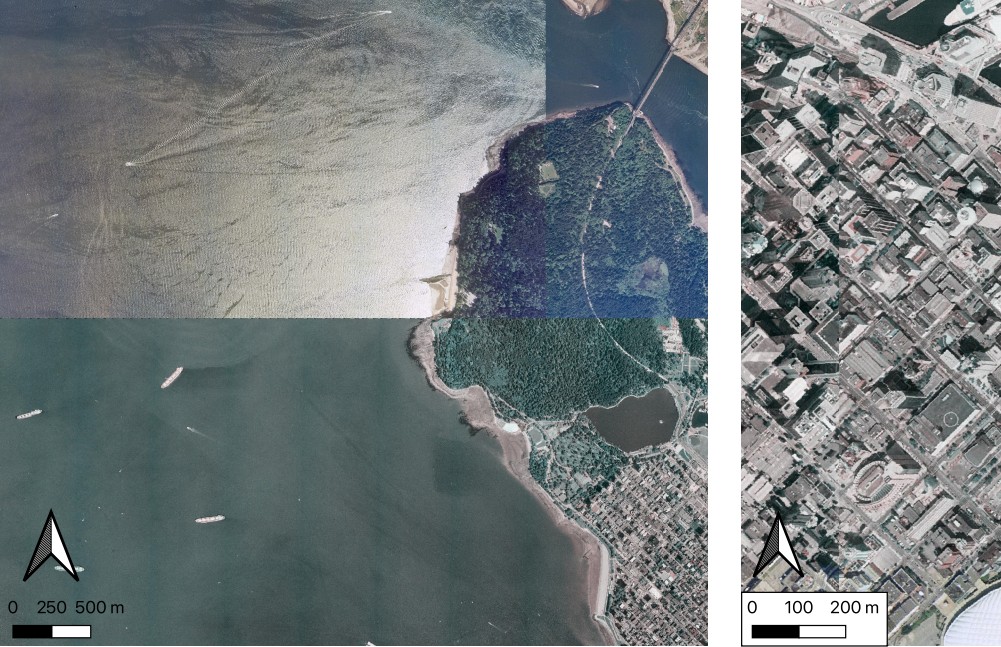

Map imagery © Google, Maxar Technologies

**Figure 2** **Examples of low quality features of RGB aerial photos showing general low-details and differences in ocean and forest color (left) and in building angles (right) across the boundaries of mosaicked aerial photos captured in 1995 over Vancouver, BC, Canada.** Map imagery Google, Maxar Technologies.

*Vancouver, 2019*). We upsampled the mask to obtain 1 m spatial resolution and reduced the 14 classes into eight classes, including water, buildings, pavement, bare, shrub, grass (including herbs), coniferous trees, and deciduous trees. We obtained a low-quality 1 m spatial resolution (RGB, 16 bit unsigned integer) aerial photos from the Canadian Wildlife Service, Pacific Region (obtained January 2022) (Fig. 2). These photos were taken over Vancouver, BC, by airplane in May and July 1995 by the Selkirk Remote Sensing Ltd. (*Selkirk Remote Sensing, 1995*) and georeferenced and othorectified by Triathlon Mapping Corporation (Burnaby, BC) (*Triathlon Mapping Corporation, 1996*).

We cropped the modern mask and RGB images to cover an area of 48,640 m × 28,672 m. More spatially extensive images will produce more training data and thus a more accurate model as long as increasing the spatial extent does not also increase the diversity of land cover types or appearances. Note that the annotated mask has limited accuracy—*e.g.*, Fig. 3A shows how it classifies the grass field in the lower left as barren. Given the limited accuracy of the original input modern land cover classification, accuracy and confusion matrices for the modern land cover should be interpreted cautiously.

To format images for training, we tiled the modern mask and modern RGB image into 512 × 512 pixels (following *Su & Chen, 2019*), resulting in 5320 tiles for each image type, where each tile of the RGB image represents the same land area as the corresponding mask

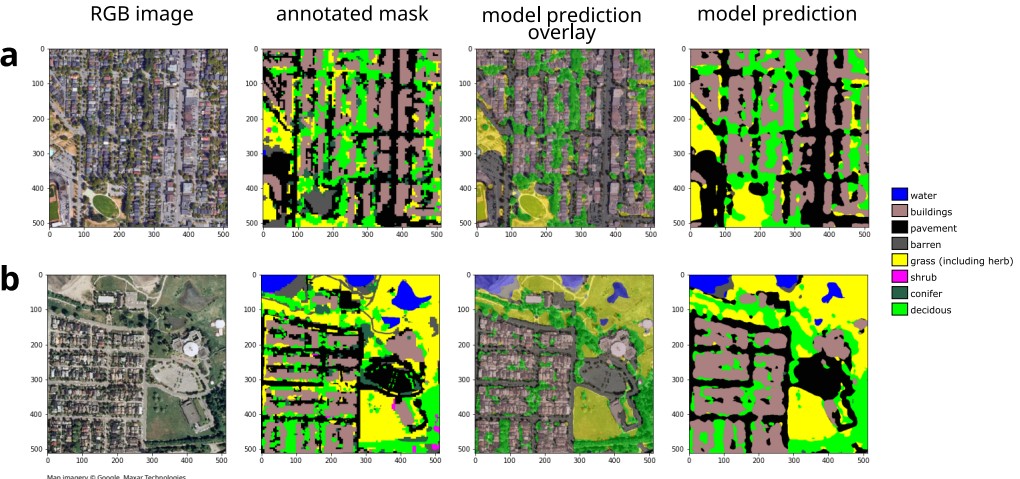

**Figure 3** **Example model inputs and land cover predictions for (A) modern images and (B) historical images showing 512 × 512 pixel tiles.** For additional example historical tile land cover classifications, see Fig. S7. Map imagery Google, Maxar Technologies.

tile (Fig. 1A). Future work might investigate the optimal tile size. We converted each tile into png format for compatibility with the TensorFlow image decoder.

## Modern deep learning model

To train the initial DeepLabv3+ model, we used TensorFlow 2 (version 2.11.0), Keras (version 2.11.0), and DeepLabv3+ within a Python (version 3.10.9) environment. Analyses were conducted on GNU Linux (6.1.12-arch1-1, 64-bit) 16 × AMD Ryzen 7 4800H processors with 15.0 GiB RAM. DeepLabv3+ can be freely downloaded and installed from https://github.com/google-research/deeplab2. We randomly sorted the mask and RGB tiles into training (90%) and validation (10%) datasets (following *Su & Chen, 2019*), then converted the tiles into tensors. Future research might test the optimality of this 90/10 ratio.

We trained the DeepLabv3+ dilated convolutional neural network on the modern tiles and masks (Fig. 1B). We used a dilation rate ranging from 1 to 18 in order to capture spatial information. Low-level features were initialized with weights from a backbone model (ResNet50 model trained on the ImageNet database; *He et al., 2015*). We trained the model for 25 epochs using a batch size of 4. We optimized the model using sparse categorical crossentropy as the loss function and the Adam algorithm for gradient-based optimization (*Kingma & Ba, 2014*).

## Modern model validation

To validate the model, we used the model to predict the land cover classes for the 10% of the modern images that were set aside for validation (Fig. 1C). We compared the resulting predictions with the original annotated masks to calculate model accuracy, produce confusion matrices, and visually inspect the accuracy of the predictions (Fig. 3A).

We modified model parameters and training dataset size until the accuracy became satisfactory.

## Historical annotations

Many deep learning analyses of land cover stop at this point (*Su & Chen, 2019*; *Du et al., 2019*). However, because our historical RGB imagery is much lower quality and was produced using different imaging technology than the modern RGB image (Fig. 2), the historical imagery likely does not represent the same feature space nor have the same distribution as the modern RGB training data. Thus following standard methods and using the model trained on the modern data to predict historical land cover classes will likely yield low accuracy (*Pan & Yang, 2010*). Instead, we manually annotated a small area of the historical image (Fig. 1D) and used transfer-learning to fine-tune the model on this small additional training dataset (Figs. 1E, 1F).

We used the ThRasE plugin (version 22.3.3a; https://github.com/SMByC/ThRasE) in QGIS (version 3.28.0) to manually annotate 58 historical RGB tiles (each tile was 512 m × 512 m, resulting in a total annotated area of $\sim$ 15 km$^2$). The first author, who carried out the annotation, spent five years in Vancouver, BC, where he conducted many avian and vegetation surveys and ran every street in the city. This first-hand knowledge of the city likely affected the manual annotations. We selected an annotation region that included coniferous forest, broadleaf forest, industrial, residential, and downtown urban core, lakes, and ocean.

To annotate the historical image, we selected an area of the image that was representative of the area of interest. We manually annotated the historical image using the same classes that were contained in the modern annotation mask. To ensure consistency between the classifications, we reviewed the relationship between the modern annotation mask and the modern RGB image. To accelerate manual annotation, the modern model can be used to create initial estimates of historical land cover cover; the modern land cover classification may also provide useful initial estimates that can then be manually corrected. QGIS and the ThRasE plugin make manual annotating easier and are freely available. Label Studio (https://labelstud.io/, accessed 20 March 2023) also provides a free platform for land cover annotation (see *e.g.*, https://www.youtube.com/watch?v=UUP_omOSKuc; accessed 20 March 2023).

After completing annotations, we tiled the historical annotation and spatially-matched historical RGB image. We doubled the size of the annotated dataset by copying each tile and rotating by 90°. Because this annotation dataset is small, such data augmentation *via* image rotation can increase model accuracy (*Perez & Wang, 2017*).

## Fine tuning

Transfer learning enables the model trained on the modern images to adapt to the lower-quality historical imagery (*Pan & Yang, 2010*). Thus, we fine-tuned the model trained on the modern images on the historical mask and image tiles (Fig. 1E) using a lower learning rate (10× lower; *Shin et al., 2016*) and only 20 epochs.

### Historical model validation

To validate the fine-tuned model for classifying the historical land cover, we used the fine-tuned model to predict the land cover class for the annotated dataset (using the entire dataset since the annotated dataset is very small; Fig. 1F). We compared the resulting predictions with the manual annotated masks to calculate the model accuracy, produce confusion matrices, and visually inspect the accuracy of the predictions (Fig. 3). To show the effect of fine tuning and data augmentation of the annotated historical images, we tested each of the three models for their capacity to estimate historical land cover, including Model 1: the original modern deep learning model, Model 2: the fine-tuned deep learning model trained without historical data augmentation, and Model 3: the fine-tuned deep learning model trained with historical data augmentation.

### Accuracy evaluation indicators

We used overall accuracy (OA) and Cohen's Kappa, $\kappa$, (Cohen, 1960) to evaluate the models, defined as:

$$accuracy = \frac{\text{number of matching cells}}{\text{number of cells}}, \tag{1}$$

$$\kappa = (p_0 - p_e)/(1 - p_e), \tag{2}$$

where $p_0$ is the observed agreement ratio between two classifiers, and $p_e$ is the expected agreement if both classifiers were random.

### Land cover prediction

To predict the land cover across the region of interest, we selected the region of interest and tiled the modern and historical RGB georeferenced images representing this region into $512 \times 512$ pixel images. We used the first model to create images predicting the land cover classes for each of the modern tiles and the fine-tuned model to create images predicting the land cover classes for each of the historical tiles. We associated the georeference information for each of the tiled RGB images with each of the predicted images and mosaicked the tiles together to create a complete modern and a complete historical land cover image. We used the final models to predict eight land cover classes for Vancouver, BC, Canada for 1995 and 2021. We corrected the land cover for water, since water is easy to manually classify. We compared the distribution of land cover classes within the city to show estimated land cover change over the 26-year period. To show more general changes, we also reduced the eight land classes into three bins: water, built (including buildings, pavement, and bare), and greenery (including shrubs, grass, coniferous trees, and deciduous trees). We used the error rate from the confusion matrices (percentage of classifications that do not match the original annotations) to plot uncertainty of land cover change.

## RESULTS

### Modern classification

Our land cover classification of the modern image produced an overall accuracy of 0.7037 relative to the 2019 Metro Vancouver land cover classification (Metro Vancouver, 2019).

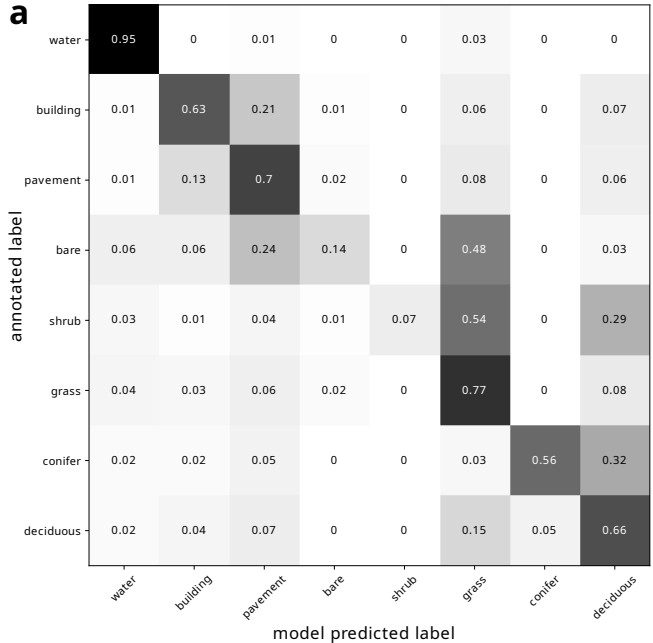
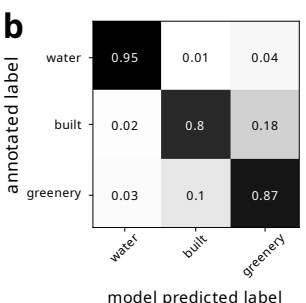

**Figure 4** Confusion matrix of modern land cover model predictions, normalized by row of (A) all land cover types and (B) binned land cover categories.

Our land cover class estimates were most consistent with the 2019 Metro Vancouver land cover classification for water (95%), and least consistent for shrubs (7%) (Fig. 4A). Binning the land cover classes into water, built, and greenery yielded improved accuracy, varying from 80% to 95% (Fig. 4B).

## Historical classification

Using the model trained only on the modern image produced low accuracy of 55.07%. It particularly struggled to classify water, mischaracterizing many non-water features as water, especially conifers (Fig. 5A). Moreover, pavement and deciduous trees were labelled incorrectly as buildings (Fig. 5A). Fine tuning the model by training on annotated historical imagery produced better predictions, improving the accuracy to 64.74%. In particular, pavement was more accurately labeled as pavement instead of mislabeled as buildings Fig. 5B). Conifers were no longer misclassified as water, though many classes were misclassified as conifers, particularly deciduous trees Fig. 5B). Water was also still overestimated, with grass in particular being misclassified as water (Fig. 5B). Fine tuning the model on *augmented* historical imagery produced much improved predictions (Fig. 5C). This final model produced an overall accuracy of 75.85% and a kappa coefficient of 68.76 relative to our manual annotation, and a clear diagonal corridor visible in the confusion matrix (Fig. 5C). This model finally correctly classified conifers, with deciduous trees only uncommonly misclassified as conifers. However, shrubs remained poorly classified. Compared to the model only trained on modern imagery, training on augmented annotated historical

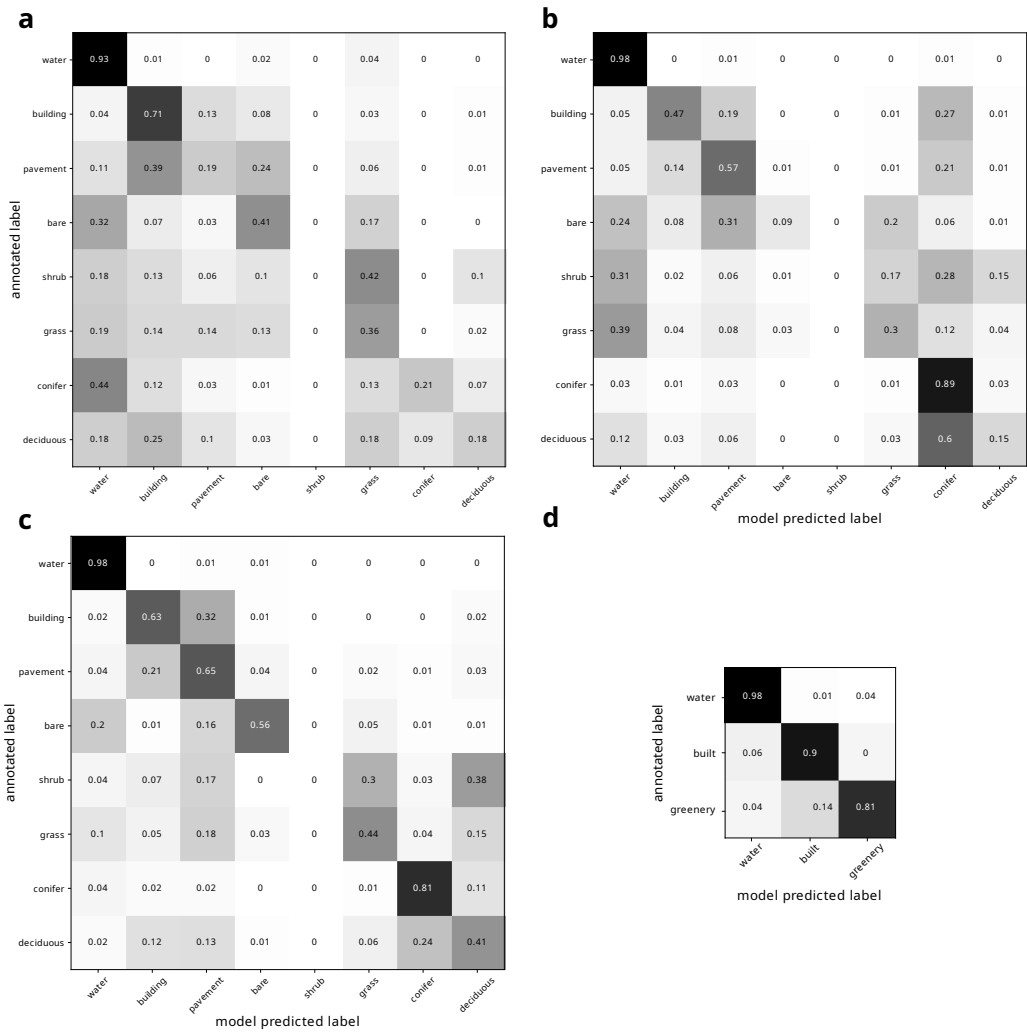

**Figure 5** **Confusion matrix of historical land cover predictions using the (A) traditional DeepLabv3+ pipeline using unaltered modern deep learning model trained only on modern imagery, (B) fine-tuned deep learning model trained on additional hand-annotated historical imagery, (C) and our pipeline presented here, using fine tuned deep learning model trained on hand-annotated historical imagery augmented by rotation.** (D) shows confusion matrix for same model as (C), but for binned land cover categories. All confusion matrices are normalized by row. The confusion matrices reveal how the models improved from (A) through (C). In (A), many landcovers were labeled as water, even when they were not water (especially conifers), and pavement and deciduous trees were labelled incorrectly as buildings. In (B), pavement was more accurately labeled as pavement rather than incorrectly labeled as buildings, though some features were still incorrectly labeled as water. However, conifers were now correctly labeled, but many (*e.g.*, grass) features that were not conifers were incorrectly labeled as conifers, especially deciduous trees and buildings. The accuracy in (C) is much improved, and the dark diagonal becomes more evident. Conifers are now correctly classified, and deciduous trees are only uncommonly misclassified as conifers. However, shrubs are still often misclassified.

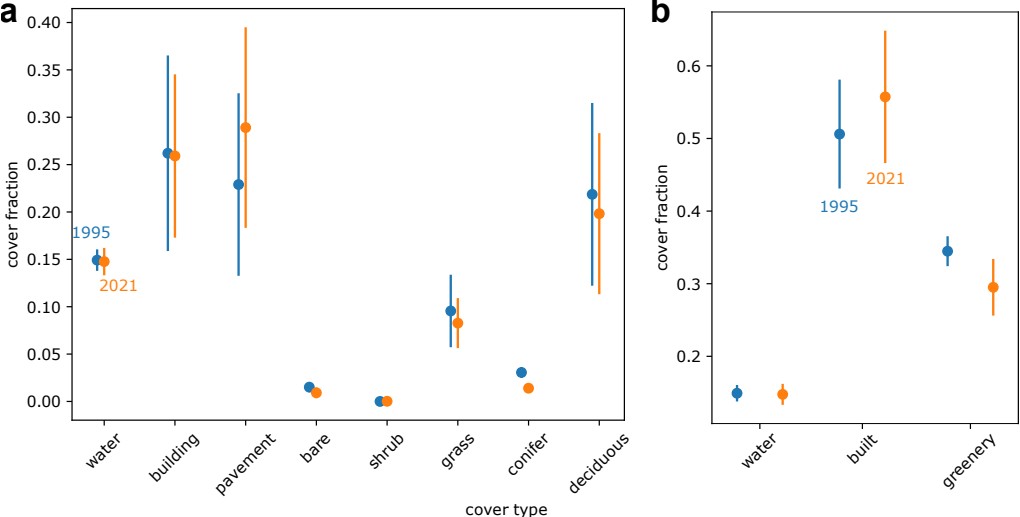

**Figure 6** **Predicted land cover change between 1995 (shown in blue) and 2021 (shown in orange) displaying (A) eight land cover classes and (B) the eight land cover classes condensed into three general bins.** Note variable y axes. Points represent estimates from deep learning model. Error bars represent deep learning model consistency with the Metro Vancouver land cover classification (*Metro Vancouver, 2019*) for 2021 estimates and with our own annotations for 1995 estimates.

imagery increased the mean intersection over union (mIoU) by nearly a factor of two (38.63% to 63.62%). Binning the land cover classes into water, built, and greenery yielded further improved accuracy (Fig. 5D). Under our architecture, all models had a computation efficiency of 6 s per step.

### Land cover change

Our land cover predictions indicate that between 1995 and 2021, conifer cover decreased by 54%, pavement cover increased by 26%, and deciduous tree cover decreased by 9% (Fig. 6A). Overall, our results indicate that greenery cover decreased by 14% and the built environment cover increased by 10% (Fig. 6B). A contingency matrix of 1995 and 2021 land cover suggests that deciduous trees, grass, and conifers may have been replaced by pavement and buildings (Fig. 7). However, some pavement (12%) and buildings (9%) may have been replaced by deciduous trees (Fig. 7).

## DISCUSSION

Our findings suggest that a deep learning network using atrous convolutional neural networks with atrous spatial pyramid pooling for semantic segmentation (DeepLabv3+) can successfully classify many land cover types from low-quality historical RGB imagery. Our pipeline harnessed modern land cover and RGB images to build an initial model, then used a small region of manually annotated and augmented historical land cover images to fine-tune this model. Our application of this pipeline to classify 1995 land cover in Vancouver, BC, Canada into eight classes yielded accuracy of over 75%. While

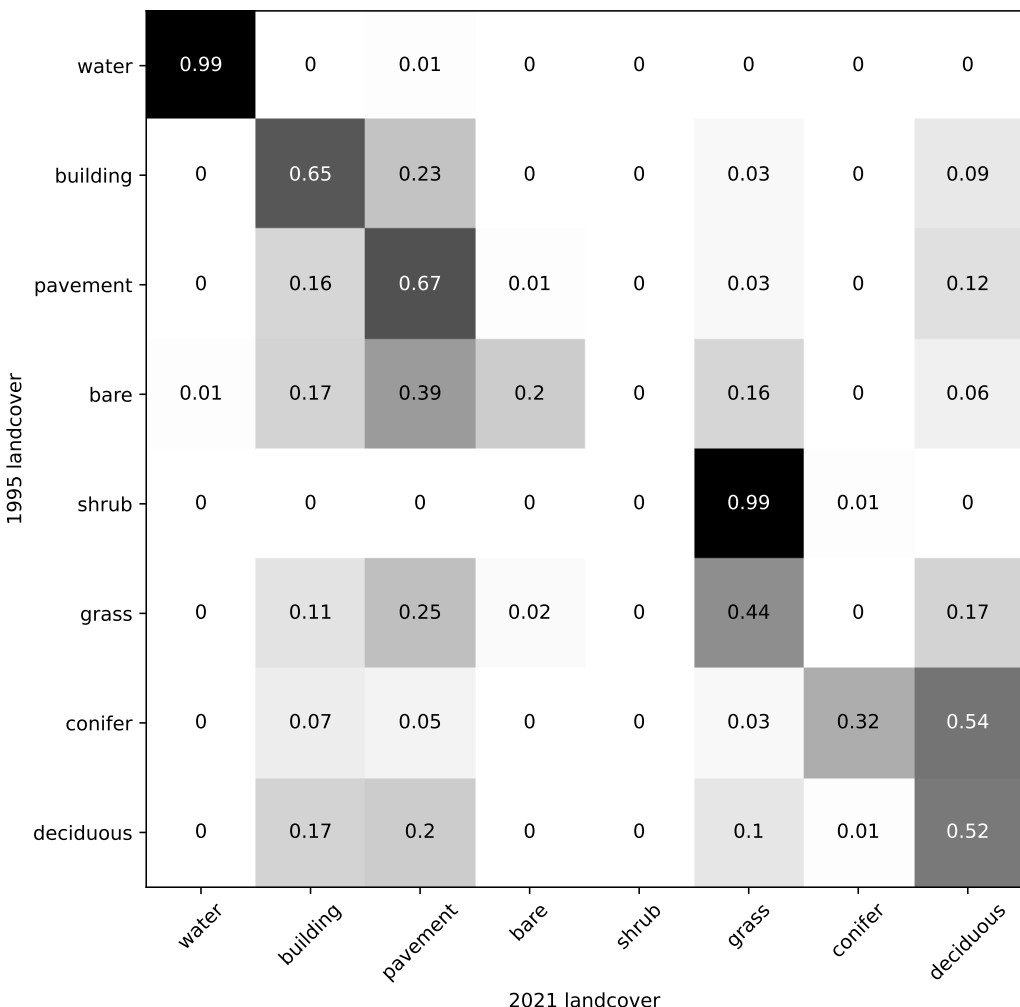

**Figure 7** **Contingency matrix showing predicted land cover change between 1995 and 2021 in Vancouver, BC.** Values normalized by column (values in each column sum to one).

DeepLabv3+'s performance at classifying modern RGB imagery has previously been demonstrated (*Du et al., 2019*), our results further show that, when combined with fine tuning and image augmentation, this deep learning architecture can successfully classify historical imagery.

Fine-tuning to historical imagery and data augmentation improved our classification in different ways. While the non-fine-tuned model incorrectly classified pavement as buildings and conifers as water, the fine-tuning was particularly helpful in enabling the model to distinguish between these land cover types (Figs. 5A, 5B), likely because the historical water imagery was particularly poor quality and buildings and pavement were more similar in the historical imagery (Fig. 2). However, the fine-tuned model still incorrectly classified many areas as coniferous. Augmenting the model helped to correctly classify conifers, likely because conifers are relatively more scarce across Vancouver's landscape, and so augmenting their representation helped to classify them.

While manual annotation can be time-consuming and inefficient, we found that manually annotating a small region of the low-quality historical image and then using these annotations to fine-tune the model produced much better results. Because this dataset is so small, augmenting the dataset with image rotation and then training on this doubled dataset further increased the accuracy of the model. A small investment in manual annotation can greatly increase model accuracy. Nevertheless, this annotation step is a disadvantage of our workflow. Additionally, the long model run-time prevented fast iterative improvement of the modeling parameters and samples.

The accuracy of our pipeline for classifying low-quality historical land cover was within the spread of previous DeepLabv3+ land cover classifications of modern imagery. Our mean intersection over union (mIoU) value of 63.62% was substantially greater than the mIoU values of 33.46–38.58% reported by *Ayhan & Kwan (2020)*. Our accuracy was slightly lower than reported by *Su & Chen (2019)* (mIoU = 75.6%). *Du et al. (2019)* achieved higher accuracy (kappa = 80% *vs* our 68.76%), but they only classified land cover into either crop area or non-crop area. Our accuracy was substantially higher than many other previous classifications, for example the DeepGlobe land cover classification challenge achieved an mIoU of only 43.3% (*Demir et al., 2018*). These comparisons suggest that our pipeline is able to distinguish land cover types in low-quality historical imagery at rates similar to other deep learning classifications of broad vegetation classes from high-quality modern imagery. Future research might test DeepLabv3+ against these more traditional algorithms, particularly for datasets that require larger models, where DeepLabv3+'s spatial pyramid pooling may not scale well due to large GPU requirements (*Chen et al., 2018*).

Our model included much higher vegetation specificity than previous models (*Ayhan & Kwan, 2020*; *Du et al., 2019*; *Su & Chen, 2019*). Such specificity of deciduous *vs* coniferous trees provides data useful for understanding Vancouver's bird diversity (*Melles, 2000*) and heat waves (*Eyster & Beckage, 2022*; *Eyster & Beckage, 2023*). Our estimate that overall greenery may have decreased, and in particular that conifer cover may have decreased (Fig. 6), suggests that the city's capacity to ameliorate urban heat waves may have decreased since 1995 (*Eyster & Beckage, 2022*).

Our deep learning model was able to classify some land cover types more accurately than others. Shrubs were particularly difficult to classify (Fig. 5B), likely due to their relative scarcity in the training dataset (shrubs was the the rarest class at only 0.015 of the modern annotated dataset). The low accuracy of shrub identification could also be due to the continuous gradient between grass–shrubs–trees, lack of height information, and the visual similarity of shrubs and grasses in the low-quality 1995 images. Some classes may be indistinguishable in low-quality images, although increasing their frequency in training datasets and more heavily penalizing rare or hard-to-classify classes (*e.g.*, through focal loss; *Lin et al., 2017*) may help. Moreover, degrading the quality of modern imagery to match historical imagery and then using these degraded imagery in concert with modern annotations to train the historical model may increase the accuracy. We expect that future work optimizing our pipeline could increase the accuracy for this and other land cover classes. Future work might also test the generalizability of our pipeline on additional time points (beyond the two we used). Moreover, note that the error rate for modern land

cover class may be over- or under-estimated, since we used the full dataset for historical validation and since the original annotation sometimes incorrectly classified land cover, while our model sometimes correctly classifies these same regions. For example, Fig. 3A shows how the original annotation (panel 2) incorrectly classifies a grass field as bare, while our model (panel 3 and 4) correctly classifies this field as grass.

## CONCLUSIONS

Historical land cover classifications are necessary for understanding land cover change. However, historical remote sensing data is often low-quality and difficult to classify. We applied a state-of-the-art deep learning algorithm, fine-tuning, and data augmentation to accurately classify low quality historical imagery. Not only does this pipeline accurately distinguish between broad land cover classes, but also fine-scale classifications of broadleaf and coniferous trees. We recommend that our pipeline might serve as a template and be widely applied to other low-quality datasets to increase our understandings of land cover change.

## ACKNOWLEDGEMENTS

Thank you to Kathleen Moore for assistance acquiring historical imagery; Roxanna Delima for feedback on figure design; and Sarah Curry for discussions that helped lead to the conceptualization of this project and for helpful comments on an earlier draft of this paper. We acknowledge that this analysis was conducted on land which has long served as a site of meeting and exchange among the Abenaki People.

### Funding

This work was supported by Environment and Climate Change Canada (GCXE22S079) and a Gund Postdoctoral Fellowship to Harold N. Eyster. The funders had no role in study design, data collection and analysis, decision to publish, or preparation of the manuscript.

### Grant Disclosures

The following grant information was disclosed by the authors:
Environment and Climate Change Canada: GCXE22S079.

### Competing Interests
The authors declare there are no competing interests.

### Author Contributions
- Harold N. Eyster conceived and designed the experiments, performed the experiments, analyzed the data, performed the computation work, prepared figures and/or tables, authored or reviewed drafts of the article, and approved the final draft.
- Brian Beckage conceived and designed the experiments, authored or reviewed drafts of the article, and approved the final draft.

## Data Availability

The code is available at OSF: Eyster, Harold N. 2024. "Code for: Applying a Deep Learning Pipeline to Classify Land Cover from Low-Quality Historical RGB Imagery." OSF. April 8. doi: 10.17605/OSF.IO/BWJKN.

## Supplemental Information

Supplemental information for this article can be found online at http://dx.doi.org/10.7717/peerj-cs.2003#supplemental-information.

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
