# Peer review of "Applying a deep learning pipeline to classify land cover from low-quality historical RGB imagery"

_PeerJ Computer Science, doi:10.7717/peerj-cs.2003_

## Round 0.1 · original submission · Major Revisions

Dear authors,
You are advised to critically respond to all comments point by point when preparing a new version of the manuscript and while preparing for the rebuttal letter. Please address all the comments/suggestions provided by the reviewers.

Please note that Reviewer 2 has suggested that you cite specific references. You are welcome to add it/them if you believe they are relevant. However, you are not required to include these citations, and if you do not include them, this will not influence my decision.

Kind regards,
PCoelho

Reviewer 1 ·

Basic reporting

This study presents land cover classifications for historical, low-quality RGB aerial imagery. The utilization of an atrous convolutional neural network from the DeepLabv3+ pipeline, trained on contemporary data and fine-tuned with historical imagery, demonstrates accurate predictions of Vancouver's land cover changes. The results indicate the effectiveness of the proposed deep-learning pipeline in classifying various land cover types. While the organization and writing are generally satisfactory, there are still some major concerns that need to be carefully clarified and revised before considering a possible publication.

Experimental design

1. In Figure 5, the accuracy of pavement in b and c has greatly improved compared to a, while the accuracy of buildings in b and c is lower than in a. Can the authors provide an explanation for this? The original modern deep learning model tended to misclassify conifer as water in 5a, but the other two models showed great improvement. It would be helpful if the authors could explain the reasons behind these improvements.

2. In lines 271–272, where the authors mention accurate fine-scale classifications of broadleaf and coniferous trees, the discrepancy observed in Figure 5b, where most deciduous trees are classified as coniferous, should be explained. Additionally, the authors should provide detailed results and discussions on the classification of broadleaf and coniferous trees.

3. The authors emphasized the importance of historical land cover and applied the newly released deep-learning method, DeepLabv3+, to assess historical, low-quality images. However, I suggest the authors compare DeepLabv3+ with one or two classic land cover classification algorithms using the same historical dataset to better understand the extent to which DeepLabv3+ has improved accuracy.

4. The authors should illustrate the computing efficiency of the three models, for example, in a table.

5. In Section 'Methods', it is suggested to add a subsection titled 'Accuracy Evaluation Indicators', encompassing overall accuracy (OA) and Kappa coefficient formulas.

6. In Line 179, for clarity, the authors can use designations such as Model 1: the original modern deep learning model, Model 2: the fine-tuned deep learning model trained without historical data augmentation, and Model 3: the fine-tuned deep learning model trained with historical data augmentation.

7. The authors may consider presenting a visual comparison of the classification results for Model 1, Model 2, and Model 3, followed by a detailed analysis. This visual comparison could illustrate how Model 3 achieves greater accuracy compared to the other two models, highlighting specific details.

Validity of the findings

1. In my opinion, the lack of innovation is one of the major issues in this study. Applying DeepLabv3+ to historical datasets, while a valid approach, does not represent a strong innovation. The authors are encouraged to elaborate on the challenges and complexities of classifying historical, low-quality image compared to high-quality images.

2. Since DeepLabv3+ falls under semantic segmentation technology, the authors should clarify in both the Introduction and Discussion why semantic segmentation is specifically advantageous for historical land cover classification. Providing this context will enhance the readers' understanding of the method's suitability for this application.

3. In the Introduction, the authors should provide a more comprehensive explanation of why they chose to apply DeepLabv3+ to historical, low-quality image classification. Are there any other studies that have attempted to apply methods used for high-resolution image classification to low-quality images, and if so, what were the results? Additionally, the differences between methods used for high-resolution image classification and traditional methods for medium to low-resolution image classification should be clarified. The authors should elaborate on the potential benefits of applying high-resolution image classification methods to low-quality images.

4. In the Discussion section, the authors should clarify why applying a deep-learning method to historical, low-quality image classification is specifically advantageous. Additionally, the authors should highlight the disadvantages of using DeepLabv3+ for historical data set classification.

Additional comments

1. Does Figure 2 indicate the entire study area? If not, it is recommended to include visualizations of the entire study area.

2. It is also recommended to include classification results from the entire study area and from a few different tiles.

3. In Figure 3, the addition of a legend indicating the correspondence between colors and categories would enhance clarity. It is suggested to ensure consistency in category names between Figure 1, Figure 3, and lines 100–101.

4. In line 188, 'W e', should be corrected to 'We'.

·

Basic reporting

The article seems to be well written, nicely designed and concise. Authors have used novel deep learning technique and explained the methodology well. In my opinion, article may be accepted after a few minor improvements.
Better to write Land cover classification as ‘land use land cover (LULC) classification’.
In abstract, kindly mention which satellite data has been used to give a better understanding for users.
Introduction is written very well and justified properly. But, I feel authors may cite some recent literatures in introduction such as: https://doi.org/10.3390/s23218966, https://doi.org/10.1080/10106049.2022.2152496, https://doi.org/10.1080/17538947.2022.2088872, etc.
The methods are well explained.
The results part seems to be weak, authors should provide more detailed results.
What about classification accuracy???
Conclusion should be enlarged. Authors should add some points for recommendations for future researchers and how the approach and techniques used in this study will help in future studies.

Experimental design

Methods are well explained.

Validity of the findings

Authors may focus on validation/accuracy checking

---

## Round 0.2 · Major Revisions

Dear authors,

Thank you for your efforts for improving the manuscript. Nevertheless, there are still some concerns to be addressed, and you are advised to critically respond to all comments point by point when preparing a new version of the manuscript and while preparing for the rebuttal letter.

Kind regards,
PCoelho

Reviewer 1 ·

Basic reporting

After carefully re-evaluating the revised manuscript, I regret to inform you that several primary concerns still require attention. Despite the authors' efforts to address them, these unresolved issues greatly affect the manuscript's overall quality and clarity.

Experimental design

Regarding my 3rd comment on Experimental Design, while the authors focus on DeepLabv3+ and its fine-tuning in this manuscript, it is crucial to assess how these models compare with classic (especially supervised) classifiers. This aspect should not be overlooked.

In response to my 7th comment on Experimental Design, while confusion metrics offer insights into accuracy, it is imperative to evaluate the performance of the three models in spatial and spectral comparison.

Validity of the findings

No comment

·

Basic reporting

Authors have comprehensively revised the MS and have addressed all the comments. I think paper may be accepted now for publication.

Experimental design

Appropriate

Validity of the findings

Very relevant

---

## Round 0.3 · accepted · Accept

Dear authors, we are pleased to verify that you meet the reviewer's valuable feedback to improve your research.

Thank you for considering PeerJ Computer Science and submitting your work.